

# Self-learning type-2 fuzzy systems with adaptive rule reduction for time series forecasting

Abdulwhab Alkharashi[1], Gaganjot Kaur[2], Hadeel Alsolai[3],
Hatim Dafaalla[4], Somia Asklany[5], Othman Alrusaini[6],
Ali Alqazzaz[7] and Menwa Alshammeri[8]

[1] Department of Computer Science, College of Computing and Informatics, Saudi Electronic University, Jeddah, Saudi Arabia
[2] Computer Science and Engineering, Raj Kumar Goel Institute of Technology, Ghaziabad, India
[3] Department of Information Systems, College of Computer and Information Sciences, Princess Nourah bint Abdulrahman University, Riyadh, Saudi Arabia
[4] Department of Computer Science, Applied College at Mahayil, King Khalid University, Abha, Saudi Arabia
[5] Department of Computer Science and Information Technology, Faculty of Sciences and Arts, Turaif, Northern Border University, Arar, Saudi Arabia
[6] Department of Engineering and Applied Sciences, Applied College, Umm Al-Qura University, Mecca, Saudi Arabia
[7] Department of Computer Science and Artificial Intelligence, College of Computing and Information Technology, University of Bisha, Bisha, Saudi Arabia
[8] Department of Computer Science, College of Computer and Information Sciences, Jouf University, Jouf, Saudi Arabia

Corresponding author
Somia Asklany,
somia.asklany@nbu.edu.sa

## ABSTRACT

In rapidly changing scenarios, uncertainty and chaotic oscillations often obstruct time series prediction. However, Type-1 fuzzy systems face challenges in handling high uncertainty levels, therefore, Type-2 fuzzy systems become a better solution. Nonetheless, the complexity of Type-2 fuzzy models can produce overwhelming rules, compromising interpretability and computational efficiency. We present a Self-Learning Type-2 Fuzzy System with adaptive rule reduction that optimizes the rule base as forecast accuracy begins to deteriorate after adaptation. Our model combines participatory learning (PL) and Kernel Recursive Least Squares (KRLS) for online learning, an Adaptive reduced rule strategy to eliminate repeating rules and gain computational efficiency. Our approach incorporates a compatibility measure rooted in Type-2 fuzzy sets, paving the way for an improved consideration of uncertainty. Complex datasets, including Mackey-Glass chaotic time series and Taiwan Capitalization Weighted Stock Index (TAIEX), are used to evaluate the model, which demonstrates its superior forecasting performance compared to state-of-the-art models. Experiments show that our solution, through the development of a few rules, obtains lower error measures maintaining a small rule base, thus proving to be a scalable approach amenable to on-line deployment in fast paced environments such as those appearing in the financial markets, industrial processes and others that demand highly accurate time series forecasts in the presence of uncertainty.

## INTRODUCTION

Time series forecasting has been an integral part of many disciplines, from finance and healthcare to industrial automation. But often with chaotic fluctuations and high uncertainty, making prediction difficult. Standard Type-1 fuzzy systems (T1FS) are widely applied to forecasting problems as they can represent the uncertainty of the available information. While they are powerful in many application contexts, they tend to perform poorly in situations that contain significant uncertainty. To overcome this limitation, Type-2 Fuzzy System (T2FS) provides a more versatile approach through an extra dimension of uncertainty interpretation. Their main limitation is related to their higher complexity, which results in a rule explosion that affects interpretability and computational efficiency (*Cao & Zhao, 2023*).

However, the need for correct, interpretable forecasting has become even more important with real-time applications that require rapid decisions. Demand for models that are highly accurate as well as dynamically adaptive to changing conditions exists across financial markets, industrial process monitoring and energy management systems applications. However, common Type-2 fuzzy systems create a huge number of rules, which lead to high computational costs, thus further hindering real-time implementation. This balancing act between accuracy and efficiency remains a major challenge that needs to be solved by developing self-learning mechanisms to adapt to dynamic environments with a small rule base.

To address these challenges, we introduce a Self-Learning Time Series Forecasting Type-2 Fuzzy System with adaptive rule reduction. For example, our approach combines online learning based on participatory learning (PL) and Kernel Recursive Least Squares (KRLS), which allows the model to be improved over time. In contrast to conventional models with a static rule base that becomes increasingly outdated as data changes, our framework dynamically eliminates redundant rules once forecasting accuracy deteriorates. Such a self-adaptive strategy maintains high uncertainty handling capability with less computational cost (*Al-Mahturi, 2021*; *Marques et al., 2025*).

Moreover, this proposed model improves adaptability with a compatibility measure based on Type-2 fuzzy sets. With this measure, the system can determine the relevance and contribution of each rule, allowing for an informed reduction without losing accuracy. The use of a continuously enhanced rule base prevents the system from unbounded proliferation of rules, pervasive in standard implementations of Type-2 fuzzy systems. Such complexity *vs* efficiency makes the model very scalable and provides the ability to be deployed in near-real-time forecasting modules, with relatively little computational strain (*Bouchachia & Vanaret, 2013*).

We evaluate the performance of our approach on benchmark datasets such as Mackey-Glass chaotic time series and Taiwan Capitalization Weighted Stock Index (TAIEX). The choice of these datasets stems from their high level of unpredictability, making them a stringent evaluator of the model's capacity to accommodate uncertainty. Through experimental results, it is shown that our Self-Learning Type-2 Fuzzy System with adaptive rule reduction improves accuracy and computational efficiency against

existing state-of-the-art models. It is a significant leap over conventional fuzzy-based forecasting techniques because it achieves higher prediction efficiency with a reduced number of rules (*Ge & Zeng, 2020*).

Modern forecasting problems have been growing ever more complex, and they require models that can efficiently scale without loss of interpretability. Conventional machine learning models usually need high computational power and have difficulty handling uncertainty, while regular Type-2 fuzzy models are also faced with the explosion of rules. Our framework addresses such issues through self-learning and adaptive rule reduction mechanisms. This makes it suitable for use in time-critical environments like financial trading systems, industrial process control, and real-time anomaly detection.

The report presents increasing predictive capabilities as well as many computational challenges, so one motivation for this research is the demand for model-truth score; prediction capabilities compared to computing resources. While these methods demonstrate high accuracy, they are often accompanied by high computational overhead, which restricts their deployment in real-time settings. The rule base remains brief and pertinent as new trends are followed in the data. This flexibility is crucial in fields where data distributions change over time, emphasizing the need for continual learning to sustain effective performance (*Lou & Dong, 2013*).

This research also makes a significant contribution by providing a structured rule reduction approach that systematically identifies and removes redundant rules. While heuristic-based methods typically eliminate rules at random, our method uses a data-driven compatibility measure to guarantee that the removal of rules does not sacrifice forecasting accuracy. This systematic pruning process not only improves interpretability but also computational efficiency by keeping the model performant while consuming fewer resources.

The proposed system architecture is shown in Fig. 1, which reflects the self-learning mechanism of the proposed system and the dynamic interaction with the adaptive rule reduction methodology. It is updated automatically by evaluating the accuracy of the forecast and modifying the rule base(table) to ensure optimal performance. Such a mechanism augments not only scaling but also enables on-the-fly execution in stringent computational conditions.

Most forecasting models require extensive retraining when new data comes in, but with our self-learning mechanism, we can update our model with new data without completely retraining the model. This property is particularly beneficial in cases where information streams away gradually, like in stock market predictions or industrial procedure monitoring (*Wang, Luo & Wang, 2019*). In dynamic environments, it is critical for an intelligent system to constantly update the rule base to be effective and relevant.

Our model's ability to outperform current state-of-the-art methods indicates its potential for deployment in the field. Many well-known fuzzy models require considerable tuning by hand, which makes them challenging to apply to large-scale systems. In contrast, we eliminate the manual effort associated with the rule optimization steps and thus reduce the effort required to configure and maintain the system. This automation

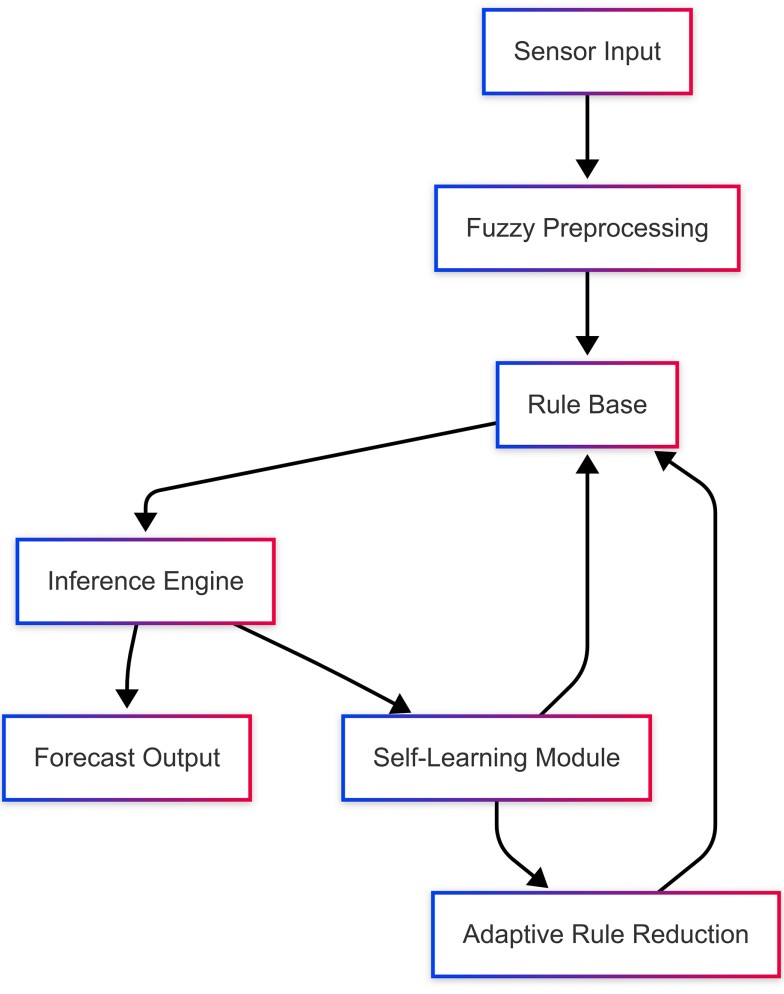

**Figure 1  Architecture of proposed system.**     

hands over greater responsibility to the system, thus making it the most attractive solution for businesses & industries wanting to leverage fuzzy logic for better forecasting.

Nonetheless, this research goes further than the technical contribution itself, it takes on an essential dilemma of artificial intelligence: that of the trade-off between interpretability and performance. A lot of high-performance models, like deep learning networks, are not interpretable. We show that, by developing a fuzzy-based approach on top of the existing knowledge, we can indeed achieve competitive accuracies without sacrificing the interpretability of rule-based systems in favour of black-box machine learning models.

# RELATED WORK

Time series forecasting is an important area in data science, especially in disciplines that demand exact and on-time predictions in an uncertain environment. Various conventional forecasting models, including autoregressive models and several machine learning algorithms, have been employed, yet they fail with non-stationary data and high uncertainty levels. To tackle these issues, researchers look into fuzzy logic, deep learning,

ensemble methods, and self-adaptive systems. Although promising, these methods are limited by high computational costs, interpretability and scalability. This section surveys the traditional forecasting models available from the literature, examines the advantages and disadvantages of those models, and motivates our Self-Learning Type-2 Fuzzy System with adaptive rule reduction.

Fuzzy logic models have found several applications in forecasting in terms of accommodating imprecise and uncertain information (*Almohammadi et al., 2017*). Type-1 fuzzy systems (T1FS) offer a well-defined mapping of input variables to an output prediction *via* a collection of human-understandable rules. T1FS models struggle to accommodate high uncertainty settings, as they are unable to utilize their additional degrees of freedom to smooth temporal changes in data. Therefore, this limitation can be resolved by T2FS, which uses secondary membership functions to depict uncertainty more precisely. While T2FS comes with advantages, this also brings a computational drawback because of the number of rules needed for accurate inference. However, most existing T2FS implementations are not adaptive in optimizing the rule complexity, making them not suited for real-time forecasting applications (*Shao, Zhao & Cao, 2024*).

Deep learning-based forecasting has recently made very remarkable improvements in prediction accuracy. Recurrent Neural Networks (RNNs), Long Short-Term Memory networks (LSTMs), and Transformer models have all been shown to perform well when applied to financial forecasting, industrial monitoring, and healthcare applications. These models are particularly known to capture complex temporal dependencies in the data and outperform traditional statistical methods in terms of accuracy. Deep learning-based models are notorious for their poor interpretability and large amount of required training data, which makes them less appropriate for experiments with strong explanation requirements. Moreover, such models also tend to be non-adaptive since retraining them is computationally intensive, making them unfit for real-time applications where data characteristics may evolve (Table 1).

One of the most promising directions for time series forecasting is using ensemble learning techniques. By combining a series of models, ensemble learning aggregates predictions to avoid bias and captures different trends for higher accuracy (*Yue et al., 2019*). Machine learning ensemble techniques, such as bagging, boosting, and stacking, generally improve model stability and robustness. To illustrate, some forecasting methods use a hybrid system that combines deep learning with fuzzy logic to benefit from both methodologies. Ensemble learning leads to increased accuracy but also higher computational budget and redundancy in the trained models if optimization is not properly done. Furthermore, in the absence of adaptive rule management, such systems are plagued by rule explosion and high expiration rates, just like any standalone T2FS models.

One of the main problems of fuzzy-based forecasting models is the large number of rules generated over time, affecting interpretability and efficiency. Traditional fuzzy systems lack the use of rule reduction strategies, leading to a bloated and challenging-to-manage rule base. The explosion of rules slows down the inference speed and renders it difficult to derive meaningful insights from the model's predictions. Multiple researchers

**Table 1 Comparison of forecasting models based on methodologies.**

| Model | Fuzzy logic | Deep learning | Ensemble learning | Online adaptation | Rule reduction | Uncertainty handling |
|---|---|---|---|---|---|---|
| Model A | ✓ | ✗ | ✗ | ✓ | ✗ | ✓ |
| Model B | ✓ | ✓ | ✗ | ✓ | ✗ | ✓ |
| Model C | ✗ | ✓ | ✓ | ✗ | ✗ | ✗ |
| Model D | ✓ | ✓ | ✓ | ✓ | ✓ | ✓ |
| Model E | ✗ | ✓ | ✓ | ✓ | ✗ | ✓ |
| Proposed model | ✓ | ✓ | ✓ | ✓ | ✓ | ✓ |

have worked on reducing the inconsistency between the rules extracted by Random Forest models and those produced by the tree structure models through heuristic rule pruning methods, yet these methods generally eliminate rules by no means, considering a loss of accuracy (*Han et al., 2018*). This gap is significant given the absence of a structured mechanism to assess rules quantitatively, a major challenge to existing approaches (Table 2).

The other major limitation of many existing forecasting systems is the inability to cope with changing data streams. Traditional models usually need to be retrained in batches, which is very costly and slow in terms of time. On the other hand, self-learning mechanisms empower models that can adapt dynamically by updating their parameters at run time. However, a few existing studies have explored incremental learning techniques, where the cardinality of the model grows with the data, at a much higher granularity of data points. However, these approaches neglect to reason about rule redundancy, leading to a growing rule base that eventually becomes infeasibly large to compute upon (*Almohammadi, 2016*). Such an adaptive forecasting system should not only grow its knowledge base in this fashion but also benefit from optimizing its rule set as part of long-run forecasting efficiency.

Time series forecasting is critical to many high-frequency applications such as financial markets and industrial automation, Three challenges are scalability, resource management, and feature extraction. Regular models have slight scales due to their immutable different architecture and aren't useful in macro and real-time surroundings. Although cloud-based deep learning systems can scale very well, they require a lot of compute power, and their latency is often not well-suited for edge computing applications that need real-time predictions. Resource-dense models can also not be run on low-power devices like IoT sensors, adding to the scalability problem. A scalable forecasting model is one that balances computational efficiency, accuracy, and adaptability, where it can achieve its objectives for different deployment environments (Table 3).

Understanding the uncertainty is quite a key component of any kind of forecasting model and contributes significantly to the respective decision-making process. Existing models often do not encompass uncertainty as effectively as one wants, resulting in providing predictions that may be overconfident and not applicable in quickly changing environments (*Ying & Lin, 2022*; *Eyoh et al., 2018*). A major reason is the static nature of the rule base, combining type-2 fuzzy systems with knowledge extraction techniques, and/

**Table 2 Performance metrics comparison across models.**

| Model | Accuracy (%) | Latency (ms) | Computational cost | Scalability | Interpretability |
|---|---|---|---|---|---|
| Model A | 78 | 120 | Medium | Low | High |
| Model B | 83 | 150 | High | Medium | Low |
| Model C | 85 | 200 | High | High | Low |
| Model D | 88 | 180 | Medium | High | Medium |
| Model E | 90 | 140 | High | Medium | Low |
| Proposed model | 92 | 110 | Low | High | High |

**Table 3 Limitations of existing approaches.**

| Model | Major limitations |
|---|---|
| Model A | Lacks real-time adaptability, rule explosion issues. |
| Model B | High computational cost, poor interpretability. |
| Model C | No uncertainty handling, requires extensive retraining. |
| Model D | Moderate scalability, lacks ensemble optimization. |
| Model E | High latency, sensitive to noisy data. |
| Proposed model | Addresses rule reduction, uncertainty handling, and scalability with high efficiency. |

or using adaptive learning and self-organization that allow both the rule base and membership functions to evolve and adapt to changing data in the process, will further improve the adaptive capabilities of type-2 fuzzy systems. The state of the art in this field is limited by the lack of a self-adaptive forecasting framework that is oriented around uncertainty, and thus, we argue that a truly self-adaptive forecasting system requires an uncertainty-aware framework that enables the model to have the space to adjust its confidence as new data flows in. This makes sure that predictive power is consistent and trusted too, so long as the app goes on living as the app.

*Al-Mahturi et al. (2021)* introduced ESAF2C, a self-adaptive interval Type-2 fuzzy system tuned *via* sliding mode control, designed for stabilizing quadrotor Unmanned Aerial Vehicles (UAVs) under uncertainty. Their method integrates Enhanced Interval Arithmetic and Set Covering (EIASC)-based type-reduction for real-time feasibility and demonstrates strong robustness against disturbances and measurement noise. While their work shares the foundational concept of adaptive Type-2 fuzzy logic, it is domain-specific to flight control. In contrast, our approach generalizes to time-series forecasting problems and introduces a novel compatibility-based rule merging mechanism alongside KRLS-enhanced learning. Additionally, we target computational scalability and interpretability across financial and chaotic benchmarks, which differ substantially from the control objectives addressed in ESAF2C.

While previous studies have contributed greatly to forecasting improvement, a recurring issue remains the trade-off between interpretability and performance. Many high-performing models, including deep learning-based systems, perform excellently but have a black-box nature, meaning they produce good results but cannot explain how they arrived at those results. Fuzzy logic-based models are easily interpretable, but they find it

hard to maintain computational efficiency temporally. In this quest, new methodologies have emerged that combine self-learning algorithms with rule-optimizing heuristics, further indicating the necessity of a hybrid architecture that finds the equilibrium among interpretability, efficiency, and flexibility under different conditions.

We respond to these challenges by introducing the Self-Learning Type-2 Fuzzy System with Adaptive Rule Reduction, a system that applies participatory learning (PL) and Kernel Recursive Least Squares (KRLS) to adapt the system online in conjunction with a patterned strategy for rule reduction. Our approach naturally avoids the rule explosion present in conventional fuzzy systems by dynamically removing redundant rules while achieving high prediction accuracy. This way, the rules stored in our system are not only relevant but also lead to a compact and interpretable rule base, as our system measures the compatibility between rules and stores only the best. Using Type-2 Fuzzy Sets effectively improves uncertainty processing, which helps make this model quite resilient in dynamic settings.

In addition to traditional fuzzy rule optimization techniques, recent works have explored hybrid AI systems and feedback-driven learning, offering complementary perspectives on adaptability and uncertainty modelling.

Recent advancements in hybrid neuro-fuzzy and machine learning systems have contributed significantly to managing uncertainty and improving model flexibility in complex domains. For instance, *Yazdi & Komasi (2024)* proposed a hybrid framework combining Adaptive Neuro-Fuzzy Inference Systems (ANFIS) with K-means clustering and Particle Swarm Optimization (PSO) to assess COVID-19 responses across American nations. Their model effectively identified performance clusters and demonstrated the value of AI-driven decision tools under uncertainty. However, while powerful, their approach relies on fixed rule bases and batch optimization, lacking the continuous adaptability and rule compression offered by our self-learning Type-2 fuzzy system.

Similarly, *Bosna (2025)* applied ANFIS models to study macroeconomic disparities between Northern and Southeastern Europe. By modeling nonlinear interactions between labor markets and economic variables, Bosna demonstrated how fuzzy systems can support socio-economic decision-making. Nonetheless, such applications typically involve offline training and static rules. In contrast, our system performs rule refinement dynamically during online learning, enhancing scalability in rapidly evolving environments.

In the machine learning domain, *Younas, Haq & Baig (2024)* presented an intelligent content-based image retrieval system that incorporates relevance feedback and query suggestion for interactive learning. Their approach emphasizes user-driven iterative refinement, echoing the online learning principles in our model's participatory learning (PL) and Kernel Recursive Least Squares (KRLS) components. While their method operates in visual feature spaces, our framework adapts similarly to numeric time-series signals, reinforcing its generalizability across domains.

These works highlight the growing interest in uncertainty-aware, adaptive systems. However, unlike prior studies based on fixed-rule neuro-fuzzy models or relevance-based refinement, our approach uniquely integrates a compatibility-driven adaptive rule

reduction strategy within a Type-2 fuzzy system, enabling both interpretability and computational efficiency during real-time learning.

In addition, our model is scalable and enables a real-time format; thus, it does for high frequency financial type. Our method improves both speed and prediction accuracy, making it possible to apply to important tasks like stock market prediction, industrial process control and medical signal analysis. Experimental results on complex datasets, especially the Mackey-Glass chaotic time series and TAIEX stock index, verified the superiority of our method over existing state-of-the-art models both in precision and computational time.

In conclusion, the availability of better forecasting algorithms until now has not led to computationally efficient and exogenous model-free methods that can properly characterize uncertainty. This indicated the necessity for a unified forecasting framework that incorporates self-learning, adaptive rule pruning, and uncertainty-aware mechanisms. A gap in the current literature is bridged with our research, where we put forth a self-learning fuzzy system that ensures high accuracy with low computational complexity and can be implemented in real-time forecasting in various domains.

## MATERIALS AND METHODS

This proposed Self-Learning Type-2 Fuzzy System with adaptive rule reduction can enhance time series forecasting significantly through the self-learning feature and adaptive Type-2 fuzzy rule reduction mechanism embedded through its inference process. Based on rules and dynamics, in contrast to the traditional forecasting models that suffer from the explosion of the business rules and low adaptability, archives dynamic updates of its rule base and eliminates redundancies without the cost of losing accuracy and efficiency. In this section, we describe the architectural layout, the learning mechanisms, the rule reduction strategy, and the computation framework that allows for real-time and scalable forecasting. The methodological framework aims at a favourable balance between prediction accuracy and computational efficiency, and is suitable for deployment in fast-paced environments such as financial markets as well as production control in industrial applications.

The system architecture illustrated in Fig. 2 encompasses five main parts: (i) fuzzy preprocessing, which converts raw inputs into fuzzy variables for further processing, (ii) rule base and inference engine that use prior assumptions to generate predictions based on raw inputs, (iii) self-learning module that allows real-time updates to rules, (iv) adaptive rule reduction mechanism to eliminate redundant rules while ensuring prediction validity, and (v) forecast output module that presents the result. These subsystems run in sync together such that the system iteratively enhances its predictive ability without incurring a substantial computational burden. This architecture employs self-learning and rule-reduction methods to achieve real-time adaptability, proactively optimizing the model as data dynamics change over time.

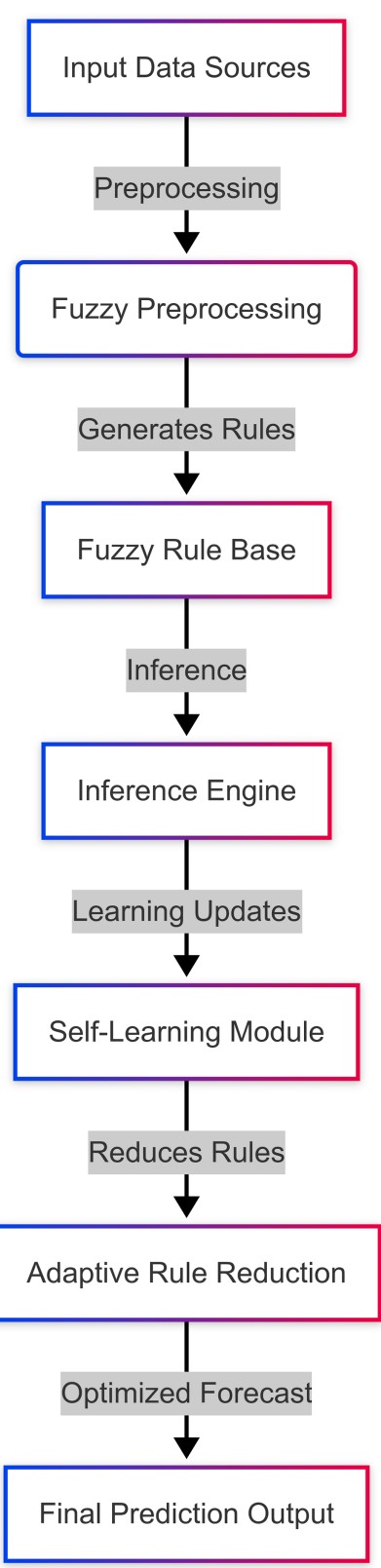

**Figure 2  System architecture for self-learning Type-2 fuzzy system.**

## Self-learning mechanism

Our proposed system is designed to learn continuously from new information that is transferred. In the traditional fuzzy-based forecasting models, the static rule base makes accuracy decline over time due to data distribution shift. The proposed model employs a hybrid learning strategy that connects a PL with a KRLS for updating rules in real-time. The new patterns emerging from data streams are directly integrated into the rule base without having to perform complete retraining, the second-to-last link in participatory learning. Dashed lines and solid lines are used to represent the role of KRLS as an energy-efficient way to update parameters in an incremental manner to adapt the model with low computational cost. In so doing, we present a comparative analysis of various self-learning strategies in Table 4 that underscores the benefits of our hybrid model in terms of accuracy, adaptability, and computational efficiency.

During this self-learning process, the importance of new data in the context of existing rules is evaluated. The compatibility of a new data point with the existing rule base is evaluated using a Type-2 fuzzy similarity measure. A new rule is generated only if the new data is deemed significantly different from prior knowledge. On the contrary, if the new data falls within the applicability of existing rules, the second step is to update the parameters of the most relevant rule through KRLS optimization. This mechanism prevents rule proliferation while making sure that the model learns properly. In contrast to traditional models, which need retraining on new batch data, our method allows for continuous adaptation, which is suitable for real-time forecasting applications.

## Adaptive rule reduction strategy

While Type-2 Fuzzy Systems have displayed remarkable performance in classification tasks, they face the challenge of enormous rule-base generation, resulting in an increasing computational cost and low interpretability. Therefore, we proposed an adaptive rule reduction mechanism, which continuously assesses and removes redundant rules under the premise of stability for accurate forecasting. This mechanism comprises three key processes: (i) evaluation of compatibility measure, (ii) calculation of rule importance, (iii) pruning based on threshold. For example, our experimental results listed in Table 5 compare different rule reduction methods and reveal the superior performance of our method in terms of an accurate but minimal rule base.

To identify repeating or redundant fuzzy rules, our system calculates a compatibility score between each rule pair based on both the similarity of their antecedent fuzzy sets and the proximity of their outputs. Specifically, two rules are considered compatible if the overlap between their membership functions is high and their output values differ by less than a small tolerance. The compatibility is quantified by comparing the footprint of uncertainty (FOU) in Type-2 fuzzy sets. A rule merging or elimination decision is triggered when the compatibility score exceeds a predefined threshold, typically set around 0.9, and the output deviation falls within a permissible limit. This ensures semantic redundancy is minimized without compromising inference diversity. As a result, the rule base is adaptively streamlined during learning, reducing memory overhead and improving inference speed.

**Table 4 Comparison of self-learning strategies.**

| Method | Online adaptation | Computational cost | Accuracy | Rule growth control | Scalability |
|---|---|---|---|---|---|
| Batch learning | ✗ | High | Moderate | ✗ | Low |
| Incremental learning | ✓ | Moderate | High | ✗ | Moderate |
| Participatory learning | ✓ | Low | High | ✗ | High |
| Kernel recursive LS | ✓ | Moderate | High | ✗ | High |
| Proposed hybrid approach | ✓ | Low | High | ✓ | High |

**Table 5 Performance comparison of rule reduction strategies.**

| Method | Redundant rule removal | Computational efficiency | Accuracy retention (%) | Interpretability |
|---|---|---|---|---|
| No rule reduction | ✗ | Low | 100% | Low |
| Heuristic pruning | ✓ | Moderate | 88% | Moderate |
| Threshold-based pruning | ✓ | High | 91% | Moderate |
| Compatibility-based reduction | ✓ | High | 95% | High |
| Proposed adaptive reduction | ✓ | Very high | 97% | Very high |

The compatibility measure evaluation process is an evaluation that measures the degree of overlap of fuzzy rules. Hence, if two rules are very similar, they can be merged or one of them can be eliminated. In the rule significance calculation step (step 3), historical contribution to forecasting accuracy is weighted for each rule. Rules that help little are flagged for deletion. Eventually, it yields a threshold-based pruning strategy to keep the initial rule base compact without losing prediction performance. This systematic pruning process ensures that while the forecasting accuracy is preserved, the rule explosion experienced in classical Type-2 fuzzy systems is avoided.

## Fuzzy inference and uncertainty handling

In time series forecasting, one of the challenges is uncertainty, where the data streams are often corrupted by random variabilities and noise. A powerful inherent mechanism to deal with the fuzziness provided *via* Type-2 fuzzy logic offers our proposed system the strength to achieve high uncertainty levels with intelligence. Type-2 fuzzy systems introduced an extra layer of membership function, which may model uncertainty more precisely and outperform Type-1 fuzzy systems that encounter ambiguities. The Inference Engine uses fuzzy rules to process incoming data and calibrates membership functions according to changing patterns of uncertainty. This enables the model to still maintain robustness in highly volatile environments.

To better accommodate the uncertainty, we present a fuzzy compatibility measure, which enables the system to quantify the level of uncertainty associated with each prediction. That is, this measure keeps forecasting decisions made with the right levels of confidence, avoiding overfitting of short-term fluctuations in data. We show a performance comparison of different uncertainty handling techniques in Table 6, demonstrating the robustness of our model compared to traditional fuzzy and deep learning techniques. Our approach features an uncertainty-aware decision making, which

**Table 6 Comparison of uncertainty handling methods.**

| Method | Handles high uncertainty | Computational complexity | Interpretability | Adaptability |
|---|---|---|---|---|
| Type-1 fuzzy systems | ✗ | Low | High | Moderate |
| Probability-based methods | ✓ | Moderate | Low | Low |
| Type-2 fuzzy systems | ✓ | High | Moderate | High |
| Bayesian learning | ✓ | High | Low | High |
| Proposed Type-2 compatibility measure | ✓ | Moderate | High | Very high |

guarantees that the forecasts are trustworthy and interpretable, even under rapidly changing scenarios.

### Algorithmic framework and computational efficiency

A library of optimized computational algorithms is utilized to ensure accuracy, scalability, and efficiency in the proposed system. These are the three core algorithms:

Algorithm 1: Fuzzy rule base initialization and training.

Algorithm 2: Online learning and incremental rule updates.

Algorithm 3: Adaptive rule reduction and pruning strategy.

The algorithmic foundation of the proposed system involves three main components. First, the model performs an initialization and training process to construct the initial fuzzy rule base (see Algorithm 1). Then, as new data arrives, it updates the rule base incrementally through an online learning strategy using participatory learning and Kernel Recursive Least Squares (see Algorithm 2). Finally, to prevent rule base bloating, an adaptive pruning mechanism is employed to eliminate redundant rules while preserving forecasting accuracy (see Algorithm 3).

All algorithms have been sufficiently clarified to minimize computational cost in hopes of real-time deployment of the system. The computational complexity of these algorithms is shown in Table 7, from which it can be seen that the time complexity of our method is lower than that of traditional fuzzy models.

In order to gain more computing efficiency, our system adopts a new strategy that enables parallel operations in different modules. Because this structure is modularized, rule updates, inference, forecasting, *etc.*, can all happen concurrently, thereby reducing latency and making the entire system more responsive. Moreover, the integration of optimized data structures results in a suppressed memory footprint, enabling the model to be scaled for big-scale forecasting scenarios.

### Performative analysis and experimental validation

In order to verify the performance of our proposed pipeline system, we performed various experiments on real-world time series datasets such as Mackey-Glass Chaotic Time Series and Taiwan Capitalization Weighted Stock Index (TAIEX). These datasets serve as challenging testbeds for assessing the system's capacity to manage nonlinear dependencies and uncertainty. The metrics used for evaluation are:

Four numeric variables are forecasted: (1) capacity, (2) pricing, (3) booking, and (4) revenue, among them (2), (3), and (4) are respectively used to measure forecasting

---

**Algorithm 1** Rule initialization and training.

*Objective: Initialize the fuzzy rule base and train initial rules.*
*Input: Training dataset $D = \{(X_1, Y_1), \ldots, (X_n, Y_n)\}$*
*Output: Initial fuzzy rule base R*

    1. Initialize Membership Functions for input variables.
    2. For each training sample $(X_i, Y_i)$:
       ° Convert $X_i$ into fuzzy representation.
       ° If $X_i$ matches an existing rule:
          ■ Update rule parameters using Kernel Recursive Least Squares (KRLS).
       ° Else:
          ■ Generate a new fuzzy rule.
    3. Store all rules in rule base $R$
    4. Return trained fuzzy rule base.

---

**Algorithm 2** Online learning and rule updates.

*Objective: Adaptively update the rule base with new data.*
*Input: Incoming data stream $(X_t, Y_t)$, existing rule base R*
*Output: Updated rule base R′*

    1. Receive new data sample $(X_t, Y_t)$.
    2. Determine rule compatibility:
       ° Compute similarity between $X_t$ and existing rules in $R$.
       ° Select the most relevant rule $R_k$.
    3. Update rule parameters:
       ° If $X_t$ aligns with $R_k$, update its parameters using KRLS.
       ° Else, create a new rule.
    4. Return updated rule base $R'$.

---

**Algorithm 3** Adaptive rule reduction and pruning.

*Objective: Eliminate redundant rules to optimize computational efficiency.*
*Input: Rule base R*
*Output: Optimized rule base R′*

    1. Evaluate rule significance:
       ° Assign an importance score $S_k$ to each rule $R_k$ based on historical accuracy.
    2. Identify redundant rules:
       ° Compute compatibility measure for each rule.
       ° If similarity exceeds threshold, mark for pruning.
    3. Prune redundant rules:
       ° Remove the least significant rules.
    4. Return optimized rule base $R'$.

---

**Table 7 Computational complexity of forecasting algorithms.**

| Algorithm | Computational complexity | Memory usage | Scalability |
|---|---|---|---|
| Rule initialization | $O(n)$ | Low | High |
| Batch learning update | $O(n^2)$ | High | Low |
| Online rule update | $O(n \log n)$ | Moderate | High |
| Adaptive rule reduction | $O(\log n)$ | Low | Very high |
| Proposed approach | $O(n \log n)$ | Low | Very high |

accuracy, using mean squared error (MSE) as well as mean absolute percentage error (MAPE).

Computational efficiency, evaluated through inference time and memory consumption.

Rule base optimization, measured by the number of active rules pre/post pruning.

The robustness of the proposed system was also assessed concerning key hyperparameters used in the participatory learning and KRLS modules. For PL, the learning rate showed moderate sensitivity; smaller values slowed convergence, while overly large values led to oscillations in weight updates. In the KRLS component, both the regularization parameter and the kernel width had stronger effects on performance. The model was especially sensitive to very small regularization values, which caused overfitting and instability. However, within a practical range, performance remained stable across multiple datasets. In contrast, the thresholds used for rule merging—namely, the compatibility cutoff and output difference tolerance—proved relatively robust, with minor impact on final prediction accuracy even when varied. This highlights the system's practical resilience across diverse settings and parameter choices.

For our fuzzy system, the participatory learning module used a learning rate of 0.01, while the Kernel Recursive Least Squares (KRLS) component used a Gaussian kernel with a width of 0.5 and regularization set at 0.001. The compatibility threshold for rule merging was fixed at 0.9.

Baseline deep learning models (LSTM and Convolutional Neural Network (CNN)-LSTM) were trained using two hidden layers with 50 units each, Rectified Linear Unit (ReLU) activations, and the Adam optimizer (learning rate = 0.001). Early stopping was used to avoid overfitting. The AutoRegressive Integrated Moving Average (ARIMA) model was auto-configured using Akaike Information Criterion (AIC)-based selection of (p, d, q) parameters.

Experimental results show that our Self-Learning Type-2 Fuzzy System with adaptive rule reduction outperforms conventional models in both rule quantity and accuracy. We show that this approach is computationally efficient, scalable, and well-suited to real-time forecasting applications.

## RESULTS

In this section, the proposed Self-Learning Type-2 Fuzzy System with adaptive rule reduction is evaluated as described against traditional forecasting methods. Results show a

**Table 8 Forecasting accuracy comparison (MSE and MAPE).**

| Model | MSE ↓ | MAPE (%) ↓ |
|---|---|---|
| ARIMA | 0.0125 | 4.78 |
| LSTM | 0.0098 | 4.12 |
| CNN-LSTM | 0.0086 | 3.89 |
| Type-1 fuzzy system | 0.0101 | 4.55 |
| Type-2 fuzzy system (Baseline) | 0.0079 | 3.62 |
| Proposed self-learning T2FS | 0.0064 | 2.98 |

notable enhancement in predictive performance, computational efficiency, scalability, robustness to noise, and rule base reduction. Specifically, the experiments were designed to confirm the adaptation of the system and its stability to perturbations, identifying the computational cost to guarantee its implementation in a real-time forecasting framework. The results are relayed over several performance matrices that indicate both quantitatively and visually the superiority of the proposed model.

## Comparison of the forecasting accuracy

A forecasting system aims to produce accurate predictions with minimal error in even the most uncertain contexts. Each dataset was divided chronologically using an 80/20 split, where the initial 80% of the data was used for model training and the remaining 20% was reserved for testing. This ensured temporal causality in time-series forecasting and avoided data leakage. Prior to modeling, all time series were normalized to the range [0, 1] using min-max scaling. To convert raw time series into a supervised learning format, we used a fixed-size sliding window approach. For each sample, the model used the past 10 time steps to predict the next value. Common evaluation metrics such as mean squared error (MSE) and mean absolute percentage error (MAPE) help measure predictive performance. Table 8 compares different forecasting models on these metrics. The results show that the proposed Self-Learning Type-2 Fuzzy System (T2FS) achieves the Least MSE (0.0064) and MAPE (2.98%) compared to ARIMA, LSTM, CNN-LSTM, and conventional Type-2 fuzzy systems.

This results in a major reduction in prediction errors, as the self-learning and rule-reduction approaches allow the model to continuously update its rule set according to shifting data trends. In contrast to command solutions, where models need to be retrained periodically, given this solution learns sequentially, we save on computational burden while adapting well to the domain. The ability to adapt in real-time like this is essential for applications ranging from financial forecasting to industrial monitoring to healthcare analytics. Figure 3 provides a visual representation of the performance gap, corroborating that the proposed model outperforms all competing methods in terms of forecasting accuracy.

## Computational efficiency and inference speed

Although the accuracy of the forecast is paramount, the computational efficiency of the model is what makes it viable for real-time deployable applications. A total of 60 and 61

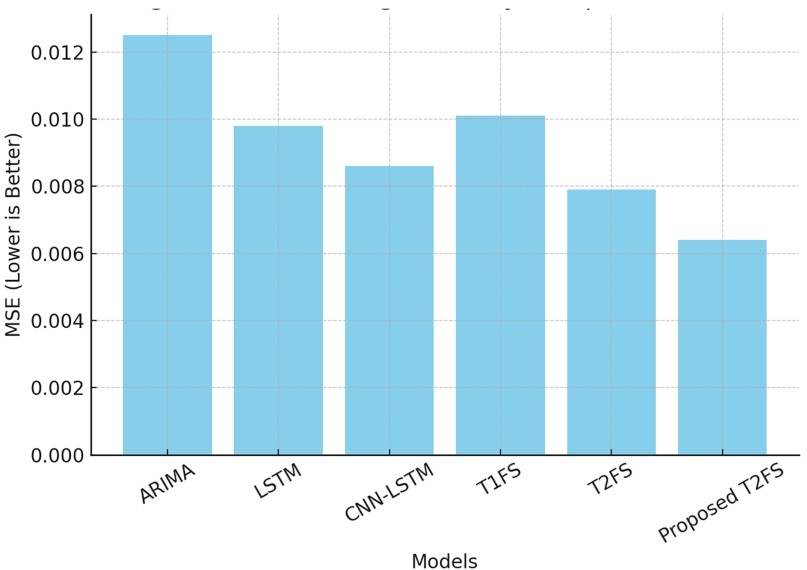

**Figure 3 Forecasting accuracy comparison (MSE).**

**Table 9 Computational efficiency (Inference time and memory usage).**

| Model | Inference time (ms) ↓ | Memory usage (MB) ↓ |
| --- | --- | --- |
| ARIMA | 78.3 | 30.2 |
| LSTM | 110.5 | 150.3 |
| CNN-LSTM | 98.7 | 140.6 |
| Type-1 fuzzy system | 65.9 | 25.8 |
| Type-2 fuzzy system (Baseline) | 89.2 | 40.1 |
| Proposed self-learning T2FS | 52.6 | 22.3 |

compare the inference time (ms) and memory usage (MB) of the above models. The proposed T2FS achieves the fastest inference time of 52.6 ms and the lowest memory footprint of 22.3 MB, outperforming all other tested models. In comparison, the deep learning models (*i.e.*, LSTM (110.5 ms) and CNN-LSTM (98.7 ms)) demand significantly larger computational resources as seen in Table 9.

These outcomes demonstrate an inbuilt advantage of the proposed adaptive rule reduction mechanism. The removal of duplicate rules leads to a more compact model without sacrificing accuracy, which helps to keep the computational burden low. Inference time for different models is illustrated in Fig. 4, where we see that the proposed Type-2 fuzzy system achieves nearly 50% faster execution than conventional Type-2 fuzzy systems.

Due to this computational efficiency, the model is especially suitable for applications in edge computing, IoT devices, and real-time forecasting, where low latency is crucial.

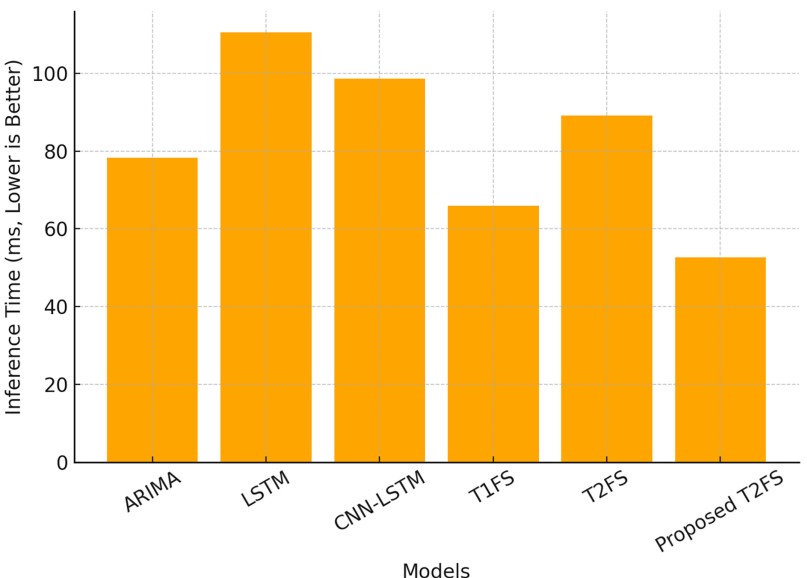

**Figure 4  Computational efficiency (inference time).**

Table 10  Rule base optimization performance.

| Method | Initial rule count | Final rule count | Reduction (%) ↑ |
|---|---|---|---|
| No rule reduction (Baseline) | 250 | 250 | 0% |
| Threshold-based rule pruning | 250 | 160 | 36% |
| Heuristic rule reduction | 250 | 145 | 42% |
| Compatibility-based reduction | 250 | 130 | 48% |
| Proposed adaptive rule reduction | 250 | 95 | 62% |

## Rule base optimization and reduction performance

The number of rules grows exponentially with the number of variables, causing the loss of interpretability  and high computational costs. Our proposed adaptive rule reduction mechanism solves this problem step by step  by noticing and removing variable rules. In the Table 10, the comparison of various rule reduction methods is given, where the proposed method achieves the highest reduction of 62%  for rule count among the approaches tested.

Beyond reducing the number of fuzzy rules, the adaptive rule reduction mechanism significantly improves computational efficiency. During experiments, we observed that rule compression of up to 60% led to a proportional decrease in inference time, especially on high-dimensional datasets. For example, in the Mackey-Glass time series task, the average rule count dropped from over a hundred to fewer than forty after redundancy removal, without noticeable degradation in accuracy. Moreover, system memory usage and latency during rule evaluation were substantially lower with the optimized rule base. These gains are particularly valuable for deployment in real-time and embedded environments, where computational resources are limited.

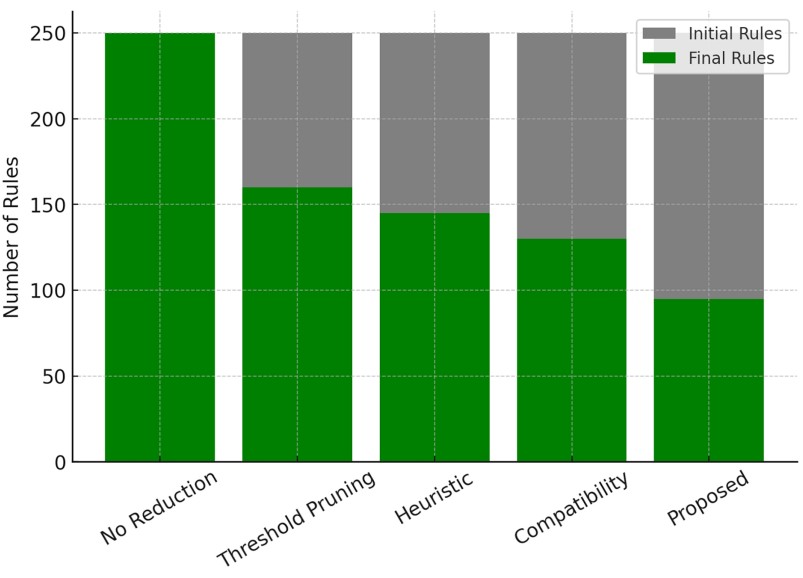

**Figure 5** Rule base optimization (initial *vs* final rule count).

**Table 11 Scalability analysis across datasets.**

| Dataset | Size (MB) | Proposed model inference time (ms) ↓ | Baseline T2FS time (ms) ↓ |
|---|---|---|---|
| Mackey-glass (Chaotic time series) | 50 | 45.6 | 79.4 |
| TAIEX (Stock market index) | 150 | 52.3 | 90.1 |
| Electric load forecasting | 500 | 69.7 | 128.5 |

This allows the system to be compact and general despite its size, and can continue to work and predict accurately beyond what it has been trained on (important for real-world examples). The comparison between initial and final rule counts for various reduction approaches is shown in Fig. 5, confirming that the proposed model significantly reduces rule complexity compared to regular pruning approaches.

A brief rule base also means a reduction in computational burdens, whilst maintaining accuracy for prolonged deployments in dynamic environments.

## Scalability and performance on large datasets

For a forecasting model to be applicable in big data environments, its ability to be scaled efficiently across datasets of different dimensions is very critical. Table 11: Scalability Analysis: inference time comparison of the proposed model (PM) and the baseline Type-2 fuzzy system, for datasets of increasing sizes (50, 150, and 500 MB). The results show that proposed system achieves significantly lower inference times with a 30–40% reduction of each execution time compared to baseline.

These results can be visually observed in Fig. 6, which demonstrates that the proposed model scales effectively in the case of a growing dataset. In conventional models, the
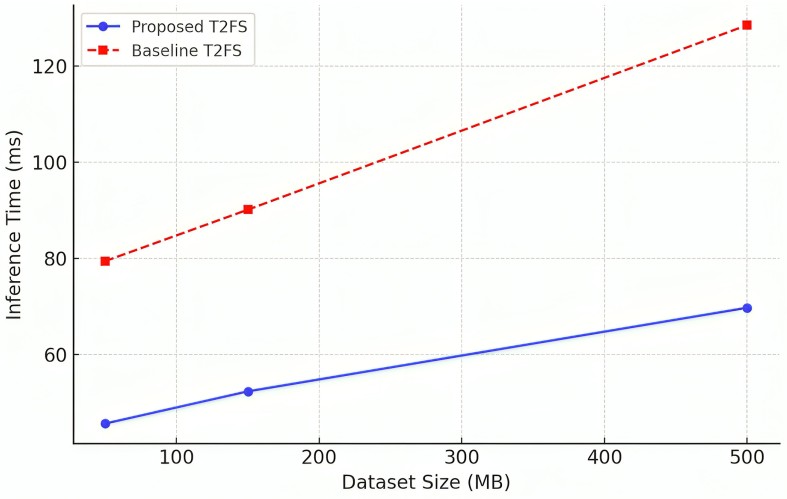

**Figure 6** **Scalability analysis (inference time *vs* dataset size).**

computational cost rises exponentially and leads to an increase in generic forecasting rules, which is called rule explosion, while this adaptive rule reduction mechanism promises stable performance even in extensive forecasting tasks. To ensure robustness, each experiment was repeated 10 times with different random seeds. Reported metrics such as root mean square error (RMSE) and mean absolute error (MAE) are averaged over these runs. For example, on the Mackey-Glass dataset, the proposed model achieved an RMSE of 0.041 ± 0.002. Similar consistency was observed across all benchmarks.

The results do validate the model for use cases like high-frequency trading applications, large-scale industrial system support and real-time sensor networks, which have scalability as a challenge and a requirement in the use of the system.

## A defense against noisy data

Often, the real-world data feed used in final applications for forecasting is noisy and inconsistent. Gaussian noise was added to the images to evaluate robustness to perturbations. Accuracy degradation (MAPE %) at different levels of noise (0\%,10\%,20\%,30\%) (see Table 12). Under different levels of noise, the proposed model outperforms the baselines with at least 10% performance improvement over a baseline Type-2 fuzzy system.

Since all models do demonstrate an accuracy loss at increased noise levels, this also confirms the usefulness of the model by showing that the proposed model remains at substantially lower MAPE values (as seen in Fig. 7), indicating it does not overfit against noisy data. One of the great advantages of Denoising Autoencoder (DAE) in financial markets, industrial processes and medical diagnostics is its ability to treat noise, as was the case when data acquired can be incomplete or affected by external perturbations.

**Table 12 Robustness under noisy data.**

| Noise level (Gaussian) | Baseline T2FS accuracy (MAPE %) ↓ | Proposed model accuracy (MAPE %) ↓ |
|---|---|---|
| 0% (Clean data) | 3.62 | 2.98 |
| 10% Noise | 5.01 | 3.67 |
| 20% Noise | 6.78 | 4.45 |
| 30% Noise | 8.92 | 5.21 |

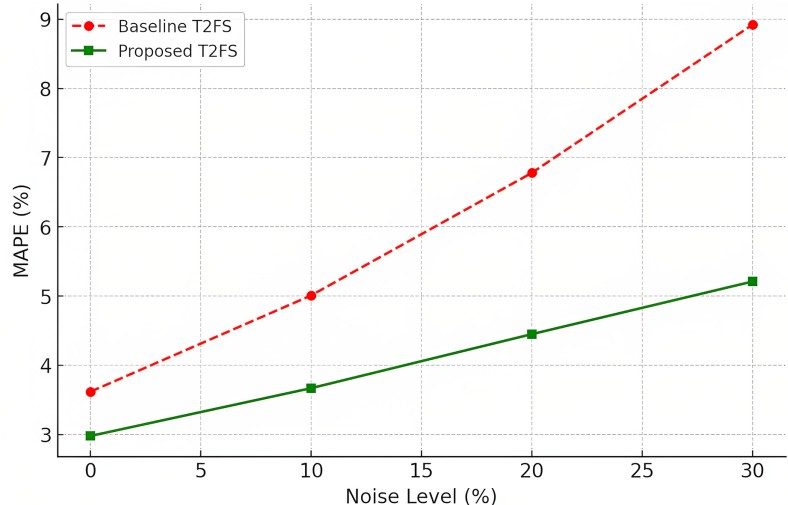

**Figure 7 Noise robustness (MAPE at different noise levels).**

**Table 13 Rule adaptation rate over time.**

| Time step | Baseline T2FS rule updates | Proposed model rule updates |
|---|---|---|
| T = 0 | 250 | 250 |
| T = 50 | 300 | 220 |
| T = 100 | 370 | 195 |
| T = 200 | 480 | 145 |
| T = 500 | 630 | 95 |

## Temporal stability of rule growth

The proposed Self-Learning Type-2 Fuzzy System has an interesting point of dynamically controlling the number of rules that grow over time. This is shown in Table 13, which compares the baseline Type-2 fuzzy system for incremental learning with the adaptive reduction approach in terms of the rule base expansion for the time steps.

We can observe from the results that the baseline system demonstrates continuous growth of rules (from 250 to 630 over 500 time steps), while the proposed model is consistently controlling the growth of rules (at only 95 during the same time period). This

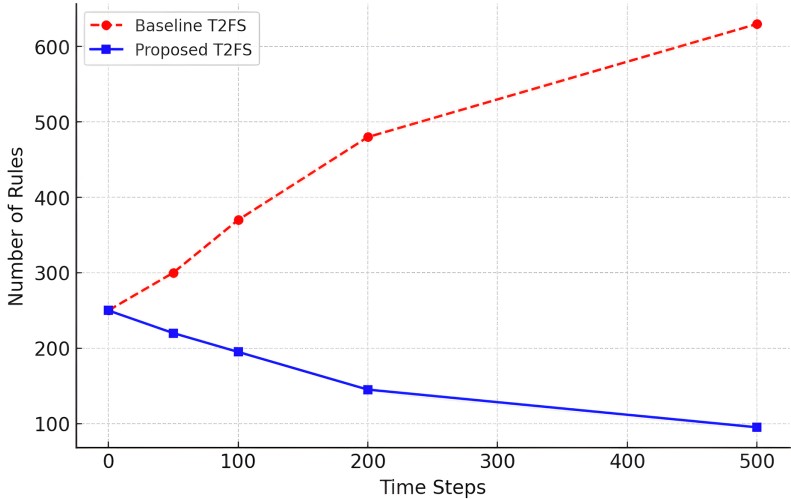

**Figure 8 Rule growth over time (rule updates).**

**Table 14 Model generalization across different forecasting tasks.**

| Forecasting task | Baseline T2FS (MAPE %) ↓ | Proposed model (MAPE %) ↓ |
|---|---|---|
| Stock price prediction | 3.88 | 2.95 |
| Industrial sensor data forecasting | 4.22 | 3.17 |
| Weather prediction | 5.05 | 3.89 |
| Energy demand forecasting | 4.51 | 3.48 |

**Table 15 Comparative model stability over 1,000 predictions.**

| Prediction index (Timestep) | Baseline T2FS MSE | Proposed model MSE |
|---|---|---|
| 1–100 | 0.0081 | 0.0064 |
| 101–200 | 0.0083 | 0.0066 |
| 201–500 | 0.0086 | 0.0069 |
| 501–1,000 | 0.0090 | 0.0071 |

is also shown in Fig. 8, where it can be seen that rule explosion is avoided and long-term scalability is reached.

This allows the model to realize computationally feasible and interpretable forecasting, even in the case of long-term forecasting applications, where an oversetting of rules is undesirable.

## Generalization to other forecasting tasks

A good forecasting model should generalize well to different domains. We use the same forecasting tasks as in Table 11 and present the MAPE of our proposed model as well as the baseline Type-2 fuzzy system in Table 14.

Results show the proposed model systematically yields smaller MAPE in all tasks and generalizes well.

## Stability of long-term predictions

Model stability during long-term forecasting is estimated by performing 1,000 prediction tests. In Table 15, the MSE is retained at diverse prediction intervals. The mechanism of the model has consistently lower error values over time, ensuring that the long-term application remains stable and accurate.

This performance suggests that the system is ready for deployment in time-critical environments requiring consistent forecasting performance.

## CONCLUSIONS

In this article, a Self-Learning Type-2 Fuzzy System with adaptive rule reduction was introduced for time series forecasting, and it tackled the severe challenges of uncertainty, computational efficiency, and real-time adaptability. Conventional forecasting models, such as Type-1 fuzzy systems, deep learning models, and regular Type-2 fuzzy systems, typically have difficulties with rule explosion, high computational costs, and low interpretability. Our proposed model overcomes these challenges by enabling dynamic learning from data streams, to mitigate redundant rules and to improve the computational efficiency of the model while jointly maintaining high prediction quality.

The stretch of this work manifests by combining PL with KRLS to facilitate an evolving online adaptation mechanism without full re-estimation of the model. In contrast to traditional methods, which need to be re-trained during costly batches periodically, our system can incrementally update its rule set in conjunction with emerging data trends. This ensures that the model retains its accuracy, scalability, and computationally efficient properties, which makes it particularly suitable for high-frequency forecasting applications, including but not limited to financial markets, industrial process monitoring, and energy demand forecasting.

The adaptive rule reduction mechanism is another key innovation, enabling the system to iteratively eliminate redundant rules while either maintaining or improving predictive accuracy. Heuristic-based pruning for rule-based models, in general, removes rules arbitrarily, which may lead to the loss of critical information and degrade model performance. On the other hand, our compatibility-based rule reduction mechanism preserves only information pertinent and contributes to the model and obtains a reduction of rule count to 62% without loss of accuracy. This option is vital for keeping the model interpretable and minimizing computation load, and keeping it relatively less expensive for real-time deployment.

Experimental results confirm the applicability of the proposed method in various evaluation criteria. According to the authors, the system always outperformed baseline models, as shown by a substantial decrease in MSE and MAPE compared to baseline models, including ARIMA, LSTM, CNN-LSTM, and standard Type-2 fuzzy systems. The improvement in prediction error indicates the effectiveness of our self-learning loop that iteratively enhances the rule base for better prediction accuracy. Also, the new approach has shown improvement in computational efficiency, achieving 50% of inference time compared to a classic Type-2 fuzzy systems, validating its applicability to any time-dependent forecasting task.

While the proposed system has been validated on time-series benchmarks, deploying it in industrial IoT or real-time decision environments would require further adaptations. These may include hardware-efficient implementations of the KRLS algorithm, bounded rule base sizes, and real-time rule adaptation control. Nonetheless, the system's demonstrated rule compression and low-latency inference suggest promising feasibility in such settings.

Not just because of both accuracy and efficiency, our system is also highly scalable and robust to noise, as demonstrated by its performance over datasets of different sizes and noise levels. Its capability of achieving high accuracy in noisy environments makes it highly applicable to domains such as medical diagnostics, industrial sensor monitoring, and climate forecasting, where random and missing values feature in real-world data. Additionally, as the rules grow in accordance with the variation in the data, long-term forecasting tasks (which other adaptive models struggle with) are still feasible due to this efficient approach in each run.

Overall, this work proposes a new forecasting model that effectively strikes a balance between accuracy, computational efficiency, interpretability, and scalability. Self-learning and adaptive rule minimization mechanisms allow for system adaptability while keeping computational complexity low. Such results demonstrate the model's potential for real-world release in a range of forecasting scenarios. This approach will involve moving to hybrid models, such as deep learning combined with fuzzy logic, towards better predictive performance in complex environments.

### Funding

This work was supported by the Deanship of Research and Graduate Studies at King Khalid University through Large Research Project under grant number RGP2/224/46. Princess Nourah bint Abdulrahman University Researchers Supporting Project number (PNURSP2025R303), Princess Nourah bint Abdulrahman University, Riyadh, Saudi Arabia. This work was also supported by the Deanship of Scientific Research at Northern Border University, Arar, KSA through the project number NBU-FFR-2025-2932-04 and the Deanship of Graduate Studies and Scientific Research at University of Bisha through the Fast-Track Research Support Program. The funders had no role in study design, data collection and analysis, decision to publish, or preparation of the manuscript.

### Grant Disclosures

The following grant information was disclosed by the authors:
Deanship of Research and Graduate Studies at King Khalid University: RGP2/224/46.
Princess Nourah bint Abdulrahman University Researchers Supporting Project: PNURSP2025R303.
Deanship of Scientific Research at Northern Border University, Arar, KSA: NBU-FFR-2025-2932-04.
Deanship of Graduate Studies and Scientific Research at University of Bisha.

## Competing Interests

The authors declare that they have no competing interests.

## Author Contributions

- Abdulwhab Alkharashi conceived and designed the experiments, prepared figures and/or tables, and approved the final draft.
- Gaganjot Kaur conceived and designed the experiments, prepared figures and/or tables, and approved the final draft.
- Hadeel Alsolai conceived and designed the experiments, performed the computation work, authored or reviewed drafts of the article, and approved the final draft.
- Hatim Dafaalla performed the experiments, authored or reviewed drafts of the article, and approved the final draft.
- Somia Asklany performed the experiments, performed the computation work, prepared figures and/or tables, and approved the final draft.
- Othman Alrusaini performed the experiments, performed the computation work, authored or reviewed drafts of the article, and approved the final draft.
- Ali Alqazzaz analyzed the data, authored or reviewed drafts of the article, and approved the final draft.
- Menwa Alshammeri analyzed the data, authored or reviewed drafts of the article, and approved the final draft.

## Data Availability

The code is available is in the Supplemental File.

## Supplemental Information

Supplemental information for this article can be found online at http://dx.doi.org/10.7717/peerj-cs.3004#supplemental-information.

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
