# Peer review of "Self-learning type-2 fuzzy systems with adaptive rule reduction for time series forecasting"

_PeerJ Computer Science, doi:10.7717/peerj-cs.3004_

## Round 0.1 · original submission · Major Revisions

Reviewer 1 ·

Basic reporting

The paper doesn't propose novel method, similar methods have been published before and the authors didn't discuss them, such as:
-Al-Mahturi, A., Santoso, F., Garratt, M. A., & Anavatti, S. G. (2021). Self-learning in aerial robotics using type-2 fuzzy systems: Case study in hovering quadrotor flight control. IEEE access, 9, 119520-119532.

Experimental design

Good

Validity of the findings

Good

Additional comments

N/A

Reviewer 2 ·

Basic reporting

The paper introduces an adaptive rule reduction strategy to improve computational efficiency. However, I miss here the exact criteria for identifying and eliminating "repeating rules". This part remained unclear.

Experimental design

My comments are as follows:
- The authors should elaborate on the specific metrics or thresholds used to determine rule redundancy
- A more detailed explanation of how the compatibility measure in Type-2 fuzzy sets influences rule reduction would strengthen the methodological transparency.

Validity of the findings

My comments are as follows:
- Discuss how this model differs from or improves upon other rule reduction techniques in fuzzy systems, such as rule merging etc. A comparative analysis of computational overhead before and after rule reduction could further highlight the efficiency gains.
- The performance of the participatory learning and kernel recursive least squares components likely depends on hyperparameter selection. Could you provide an analysis of how sensitive the model is to key hyperparameters, such as learning rates or kernel parameters? This would help readers understand the robustness of the approach in different settings.

Additional comments

My comments are as follows:
- The experiments focus on chaotic time series and stock indices, but the broader applicability of the method is only briefly mentioned. The authors should discuss potential challenges or adaptations needed to deploy the model in other fast-paced environments, such as industrial IoT or real-time decision systems?
- The literature review should be extended to adequately cover Type-1 and Type-2 fuzzy systems. It should also include recent advancements in hybrid neuro-fuzzy systems and deep learning-based uncertainty quantification methods. I suggest authors to read the below interesting papers: ** for neuro fuzzy models (Yazdi and Komasi. (2024). Best Practice Performance of COVID-19 in America continent with Artificial Intelligence. Spectrum of Operational Research) **for ML techniques (Younas et al (2024). A Framework for Extensive Content-Based Image Retrieval System Incorporating Relevance Feedback and Query Suggestion. Spectrum of Operational Research) ** fuzzy logic systems (Bosna, J. (2025). Examining regional economic differences in Europe: The power of ANFIS analysis. Journal of Decision Analytics and Intelligent Computing)
Incorporating a discussion on how these approaches compare to your method in terms of uncertainty handling and rule complexity would provide a more comprehensive background and highlight the novelty of your contribution.

Reviewer 3 ·

Basic reporting

The manuscript has numerous grammatical errors, awkward phrases, and some unclear or colloquial expressions.
Example: “our system archives dynamic updates” should be “achieves dynamic updates.”
“So long as the app goes on living as the app” is unclear and non-academic.

Sentence structures need tightening to maintain clarity and professionalism throughout.

Although the text references multiple figures (e.g., Figure 3, Figure 7) and tables (e.g., Table 10, Table 14), they are not included in the submission file.

This violates basic reporting standards and makes it difficult to verify the claims.

Experimental design

The manuscript does not mention how data were divided (e.g., training/test split, cross-validation).

There is no mention of preprocessing steps (normalisation, lag creation, etc.), which are vital in time series forecasting.

There is no clear description of the fuzzy system's hyperparameters or comparative baselines like LSTM or ARIMA. Please add a table to compare them.

Tables and figures mentioned in the text (e.g., Table 9, Figure 4) are not included.

Validity of the findings

No confidence intervals, p-values, or variance/error bars are provided for the reported performance metrics.

---

## Round 0.2 · accepted · Accept

The paper can be accepted. Congratulations.

Reviewer 2 ·

Basic reporting

All the reviewers' comments have been addressed carefully and sufficiently. The revisions are rational from my point of view. I think the current version of the paper can be accepted.

Experimental design

No comment.

Validity of the findings

No comment.

Additional comments

No comment.

Reviewer 3 ·

Basic reporting

The authors readdressed all issues.

Experimental design

The authors readdressed all issues.

Validity of the findings

The authors readdressed all issues.